# Integrated Metabolomic and Transcriptomic Analyses of the Flavonoid Biosynthetic Pathway in Relation to Color Mutation in Roses

**DOI:** 10.3390/biology14101337

**Published:** 2025-09-29

**Authors:** Yun Xuan, Jie Ren, Zhu Chen, Dan Shi

**Affiliations:** Laboratory of Landscape Engineering, Institute of Agricultural Machinery Equipment and Engineering, Anhui Academy of Agricultural Sciences, Hefei 230031, China; renjieaaas@sina.com (J.R.); chenzhu0621@126.com (Z.C.); windman208@163.com (D.S.)

**Keywords:** rose, mutation, petal color, metabolomic, transcriptomic, anthocyanin biosynthesis, flavonoids

## Abstract

The red petals of the rose cultivar ‘Silk Road’ (SR) and the white petals of its color mutant ‘Arctic Road’ (AR) were examined in this study. In the blooming flowers of AR and SR, 479 flavonoid-related metabolites and 39,201 genes were identified. Comparative analyses revealed significant differences in 277 metabolites and 2556 genes between AR and SR. The contents of 11 anthocyanins, 11 proanthocyanidins, as well as the expression levels of *CHS*, *ANS*, *3GT*, *COMT*, and *CCoAOMT* differ significantly between the two cultivars, which may contribute to the formation of white petals in AR. Furthermore, 5 *GSTs*, 4 *ABCCs*, and 8 *MATEs* exhibited significant downregulation in AR. These genes may lead to weak sequestration of anthocyanins in petal vacuoles. Additionally, *Chr1g0360311* (*MYB*) may play a key role in participating in anthocyanin biosynthesis.

## 1. Introduction

As highly ornamental plants, roses are widely grown throughout the world. They not only have ornamental value but also exhibit edibility and medicinal properties, and can also be used in cosmetics [1,2,3]. Flower color acts as a pivotal trait of ornamental roses. Anthocyanins endow flowers with hues spanning from pale pink to purple, whereas carotenoids primarily produce yellow colors [4]. With the exception of green rose flowers, the main pigments responsible for the diverse coloration in roses are anthocyanins and carotenoids [5]. Red and pink roses derive their color predominantly from anthocyanins [6]. Anthocyanins play vital roles in protecting plants from biological damage and attracting pollinating insects [7,8]. In particular, anthocyanins can protect plants against high- intensity light, help them respond to biotic and abiotic stresses, and efficiently scavenge oxygen free radicals [9,10,11,12].

Moreover, anthocyanins provide significant nutritional and health benefits, particularly in terms of disease-preventive effects and antioxidant properties [13,14,15,16]. Recent studies have demonstrated that anthocyanins can improve vision in patients with glaucoma [17], assist in the treatment of retinal disorders [18], help alleviate memory impairments [19], lower blood lipid levels, reduce cholesterol [20], and contribute to the prevention and management of cardiovascular diseases [21].

The anthocyanidins present in rose petals are predominantly composed of three types: pelargonidin, cyanidin, and peonidin [22]. Biolley and Jay [23] demonstrated that color variation in modern roses is closely correlated with the contents of cyanidin 3,5-diglucoside (Cy3G5G) and pelargonidin 3,5-diglucoside (Pg3G5G). Schmitzer et al. [24] further revealed that, in addition to the predominance of pelargonidin and cyanidin 3,5-diglucosides as major pigments in rose cultivars, their 3*-O-*glucoside derivatives also play a critical role in floral pigmentation. Notably, certain varieties have been found to contain peonidin 3*-O-*glucoside (Pn3G). Moreover, novel anthocyanin glycoside structures—such as cyanidin and peonidin 3*-O-*rutinosides, cyanidin and peonidin 3-*p*- coumaroylglucoside-5*-O-*glucosides, and cyanidin 3*-O-*rhamnoside—have been identified in wild rose species and specific cultivated varieties [25,26]. In recent years, a total of twenty-two anthocyanin components have been identified, primarily comprising cyanidin, pelargonidin, peonidin, delphinidin, and petunidin, which are preferentially enriched in pink and black-red petals [6].

The biosynthesis of anthocyanins in plants is a specialized branch of the flavonoid synthetic pathway, which falls under the broader category of secondary metabolite production [27]. Phenylalanine acts as the primary precursor for anthocyanin biosynthesis. The conversion of phenylalanine into anthocyanin glycosides typically involves three distinct phases. In the first phase, the initial reaction of flavonoid metabolism occurs. The precursor substance, phenylalanine, is transformed into 4-coumarate CoA via the sequential actions of phenylalanine ammonialyase (PAL), cinnamate 4-hydroxylase (C4H), and 4-coumarate CoA ligase (4CL). In the second phase, important reactions in flavonoid metabolism occur. 4-Coumaroyl-CoA and malonyl-CoA are acted upon by chalcone synthase (CHS) to generate chalcone. Subsequently, chalcone is isomerized by chalcone isomerase (CHI), resulting in the formation of naringenin. Subsequently, naringenin is converted into dihydroflavonols through the enzymatic action of flavanone 3-hydroxylase (F3H), flavonoid 3′-hydroxylase (F3′H), and flavonoid 3′,5′-hydroxylase (F3′5′H). Finally, leucoanthocyanidin is synthesized when dihydrokaempferol and dihydroquercetin undergo catalytic conversion by the enzyme dihydroflavonol 4-reductase (DFR). The third phase involves the synthesis of diverse anthocyanin glycosides. Leucoanthocyanidins are oxidized to their colored counterparts by anthocyanin synthase (ANS). Subsequently, different colored anthocyanin glycosides are formed through the catalytic action of glycosyltransferases (GTs), methyltransferases (MTs), and acyltransferases (ATs).

CHS acts as a key enzyme in the flavonoid biosynthetic pathway. Downregulation of CHS gene expression has been shown to alter the flower color of petunias, transitioning from purple to white [28]. ANS is a critical terminal enzyme in the plant anthocyanin biosynthetic pathway. Its main biochemical role is to catalyze the conversion of leucoanthocyanins into colored anthocyanidins, which are fundamental pigments responsible for flower coloration [29].

Flavonoids, as key secondary metabolites in plants, are synthesized via a pathway that is coordinately regulated by structural genes and regulatory genes [30]. Initially, flavonoids are unstable; they are subsequently converted into stable flavonoid derivatives through sequential modifications. The final flavonoid derivatives accumulate in the vacuole [31]. There are three transport modes for the transport of flavonoids from the cytoplasm to the vacuole: namely, glutathione S-transferase (GST)-mediated transport, membrane transport proteins, and vesicle-mediated transport [31,32].

In plants, the biosynthesis of anthocyanins is principally regulated by the MYB-bHLH-WD40 (MBW) complex [33,34,35,36]. Moreover, MYB transcription factors serve as critical regulators in anthocyanin biosynthesis [35,36,37]. In addition, the bZIP family member HY5 has been confirmed to participate in the regulation of anthocyanin biosynthesis [38,39,40]. WRKY and MADS-box transcription factors may also be involved in regulating anthocyanin metabolic pathways [29,41]. In rose, *RcMYB1* and *RcMYB114* participate in regulating anthocyanin biosynthesis [36,37]. *RrMYB114*, *RrMYB108*, and *RrC1* in *Rosa rugosa* may play critical roles in regulating petal color through the modulation of multiple structural gene expressions [42]. *RhF3′H* and *RhGT74F2* were functionally validated through transient overexpression assays, confirming their involvement in anthocyanin accumulation in the ‘Rhapsody in Blue’ rose [43]. Previous studies have investigated the mechanisms underlying rose coloration. However, the molecular mechanisms controlling anthocyanin biosynthesis in roses remain unclear. In this study, we utilized a rose cultivar, ‘Silk Road’, and its color mutant, ‘Arctic Road’, to investigate the molecular mechanisms influencing anthocyanin content in their flowers.

## 2. Material and Methods

### 2.1. Plant Materials

The rose cultivar ‘Silk Road’ (China, ‘Sichouzhilu’) and its natural bud mutant ‘Arctic Road’ were grown in containers at the Anhui Academy of Agricultural Sciences in Hefei, China (31°58′ N, 117°25′ E). Rose petals were collected from fully bloomed AR and SR flowers in April 2025. Petal samples gathered from six flowers were pooled to form a single composite sample. Three independent biological replicates were employed for sample collection, and all samples were promptly frozen in liquid nitrogen, then stored at −80 °C until subsequent analysis.

### 2.2. Extraction and Comparative Quantitative Analysis of Flavonoid Metabolites

Petal samples underwent vacuum freeze-drying, and were then pulverized into powder using a grinder (MM 400, Retsch, Haan, Germany). The grinding was carried out at a frequency of 30 Hz for 1.5 min. Thirty milligrams of the accurately weighed powder was then dissolved in 1500 μL of a pre-cooled 70% methanol aqueous solution containing an internal standard. After centrifugation at 12,000× *g* for 3 min, the supernatant was filtered through a 0.22 μm microporous membrane to remove particulates, and then collected into an injection vial for subsequent analysis.

All sample extracts underwent analysis using a UPLC-ESI-MS/MS system (UPLC, ExionLC™ AD, AB SCIEX, Singapore; https://sciex.com.cn/). The effluent was alternatively connected to an ESI-triple quadrupole-linear ion trap (QTRAP)-MS. For the qualitative analysis of metabolites, it was implemented in accordance with the secondary spectrum information, relying on the self-established Metware database (MWDB). For the analysis of AR and SR, differentially accumulated metabolites (DAMs) in this study were identified by the criteria of VIP > 1 and an absolute Log_2_FC ≥ 1.0. The prcomp function in R (www.r-project.org, version 4.1.2) was utilized to perform PCA.

### 2.3. RNA Extraction and cDNA Library Construction

RNA was extracted from the petals using a combination of the CTAB method and the pBIOZOL reagent (BSC55M1, Bioer Technology, Hangzhou, China). The ground flower petal samples were mixed with 1 mL of CTAB-pBIOZOL reagent and incubated in a constant temperature mixer at 800 rpm for 10 min at 65 °C. The samples were then centrifuged at 12,000× *g* for 5 min at 4 °C. The supernatant was transferred to 200 μL of chloroform and mixed thoroughly. Following another centrifugation at 12,000× *g* for 5 min at 4 °C, the aqueous phase was collected and added to the corresponding wells of the reagent plate containing DNase I and DNase I Reaction Buffer. The reagent plate was subsequently loaded into a fully automatic nucleic acid extraction and purification instrument (Vazyme Biotech, VNP-32P, Nanjing, China). Upon completion of the instrument protocol, the purified RNA was collected. Subsequently, the extracted RNA was accurately quantified using a Qubit fluorometer (Thermo Fisher Scientific, Waltham, MA, USA). Additionally, the RNA integrity number (RQN value) was determined using a Qsep400 high-throughput biofragment analyzer (Guangding Biotech, Taiwan, China).

Following the extraction of total RNA, Oligo(dT) magnetic beads were employed to enrich mRNA. Afterward, a fragmentation buffer was used to cleave the enriched mRNA into shorter fragments. Subsequently, reverse transcription was performed using random primers to generate double-stranded cDNA. The obtained cDNA fragments were purified, and then underwent end repair, dA-tailing, and adapter ligation. Finally, the library was amplified by means of phi29 DNA polymerase to produce DNA nanoballs (DNBs). Each of these DNBs contained over 300 copies of the original molecule. The DNBs were then loaded onto the sequencing chip and subjected to sequencing on the MGI sequencing platform (MGI Tech, Shenzhen, China).

### 2.4. RNA-Seq Analysis

Raw sequencing data underwent filtering via fastp to eliminate adapter-containing reads, those with over 10% unknown nucleotides (N), and low-quality reads (Q-value ≤ 20) accounting for over 50% of the read length. The resulting clean reads were used for subsequent analyses. The reference genome along with the corresponding annotation files was retrieved from a specified website (https://www.rosaceae.org/analysis/282, accessed on 13 June 2022). Feature Counts was employed to compute gene alignment statistics, and then the FPKM (Fragments per Kilobase of transcript per Million fragments mapped) value for each gene was calculated according to their respective gene lengths. A threshold of FPKM ≥ 1 in at least one sample was set to define expressed genes. The differential gene expression between the two groups was analyzed using DESeq2 (version 1.22.1), and the *p*-values were adjusted using the Benjamini & Hochberg method. Differentially expressed genes were determined as those with a minimum fold change (FC) of 2 and an adjusted *p*-value below 0.05. Next, gene annotation was performed, followed by GO enrichment analysis and KEGG pathway enrichment analysis.

### 2.5. Association Analysis of Metabolomic and Transcriptomic Data

To comprehensively analyze the transcriptomic and metabolomic datasets, we calculated the Pearson correlation coefficients. Statistically significant correlations were defined as those with a coefficient |r| > 0.9 and a *p*-value < 0.01. Transcriptomic and metabolomic data collected from both the AR and SR groups were combined for gene–metabolite network analysis. Ultimately, the associations between transcripts and metabolites were visualized using R (igraph, version 1.3.4).

## 3. Results

### 3.1. Quantitative Analysis of Rose Petal Metabolites Between AR and SR

The ‘Silk Road’ (SR) rose exhibits red flowers. Its natural bud mutant variety, ‘Arctic Road’ (AR), displays white petals with a faint reddish hue on the abaxial surface of the outer petals (Figure 1). To investigate differences in flavonoid composition, rose petal samples underwent UPLC-ESI-MS/MS analysis using an extensive targeted metabolomics approach.

Principal Component Analysis (PCA) was conducted to identify flavonoid metabolic differences between groups as well as variations among intra-group samples. As shown in Figure 2A, the PCA score plot shows a distinct separation between the SR and AR groups (PC1 = 77.11%, PC2 = 8.6%). A heatmap (Figure 2B) illustrates the profiles of 479 identified metabolites in the rose petals of the two groups. Overall, these findings indicate substantial differences in the accumulation patterns of flavonoid metabolites between SR and AR, which may underlie the observed phenotypic variations.

In total, the identified metabolites were categorized into 12 functional classes (Appendix A). Detailed classifications indicate that these metabolites comprised 12 anthocyanins, 18 proanthocyanidins, 183 flavonols, 104 flavones, 24 flavanones, 47 tannins, 40 flavanols, 12 aurones, 15 chalcones, 10 isoflavones, 4 flavanonols, and 10 other flavonoid derivatives. Furthermore, between the AR and SR groups, 277 differentially accumulated metabolites (DAMs) were detected in total. When compared to the SR group, in the AR group, 124 metabolites were downregulated, and 153 metabolites were upregulated (Table 1, Table 2 and Appendix A). These DAMs could be divided into twelve categories, with the majority belonging to five main classes: flavonols (45.5%), flavones (24.5%), flavanols (5.78%), anthocyanins (3.97%), and proanthocyanidins (3.97%) (Table 1). Additionally, the majority of flavonols were found to have higher concentrations in AR compared to SR, and a similar trend was observed for flavones.

In contrast, 11 anthocyanins and 11 proanthocyanidins exhibited significantly reduced levels in AR (Table 2). These 11 anthocyanins include cyanidin 3,5*-O-*diglucoside, cyanidin 3*-O-*beta-D-sambubioside, cyanidin-3*-O-*galloyl-galactoside, cyanidin-3- diglucoside-5-glucoside, cyanidin 3,3′,5-tri*-O-*glucoside, cyanidin-3*-O-*(6″*-O-*feruloyl) glucoside, cyanidin 3*-O-*(beta-D-xylosyl-(1→2)-beta-d-galactoside), peonidin 3*-O-*glucoside, peonidin-3,5*-O-*diglucoside, peonidin 3*-O-*sophoroside, and pelargonidin 3,5-di-beta- d-glucoside. These findings suggest that the significant reduction in these anthocyanins may directly result in the development of white coloration in AR flowers. Furthermore, quercetin and its multiple derivatives were detected in AR and SR. Among them, 17 compounds showed a decrease and 14 compounds exhibited an increase in AR (Appendix A). Additionally, kaempferol and its numerous derivatives were also detected. Among them, 13 compounds demonstrated a decrease, and 47 compounds showed an increase in AR (Appendix A).

### 3.2. Transcriptome Sequencing and Analysis

Through RNA-seq analysis, a total of 352,814,168 paired-end clean reads were obtained, including 168,475,584 reads from AR samples and 184,338,584 reads from SR samples (Appendix A). Approximately 52.9 Gb of clean data were generated in total, with each sample contributing an average of around 8.82 Gb. The Q30 value was over 94.69%, and the average GC content was approx. 45.7% (Appendix A). Significantly, in all six samples, more than 88.7% of the reads were successfully mapped to the reference genome of *Rosa chinensis* (Appendix A). These results indicate that the vast majority of sequencing reads can be mapped to corresponding positions in the reference genome, implying that these rose cultivars exhibit a high degree of genetic similarity to the reference genome.

### 3.3. Identification of Differentially Expressed Genes (DEGs) in the Petals of AR and SR

In both AR and SR samples, 39,201 genes were detected in total. The FPKM method was employed to quantify their expression levels (Appendix A). A comprehensive comparative analysis was conducted to explore the DEGs associated with significant changes in petal color (AR vs. SR). In this comparison, there were 2556 specific DEGs, among which 1261 were upregulated and 1295 were downregulated (Figure 3A). The volcano plot illustrates the overall distribution of 20,369 genes between the AR and SR groups, with red dots representing upregulated DEGs, blue dots denoting downregulated DEGs, and grey dots signifying non-significantly differentially expressed genes (Figure 3B). Furthermore, a heatmap displaying the expression profiles of the 2556 DEGs in rose petals from the AR and SR groups is presented in Figure 3C. These results suggest that the number of upregulated DEGs is comparable to that of downregulated DEGs in the comparison between AR and SR.

To investigate the genes associated with the difference in petal color between AR and SR, Gene Ontology (GO) and Kyoto Encyclopedia of Genes and Genomes (KEGG) pathways analyses were performed on the DEGs. A total of 2556 DEGs identified in the AR vs. SR comparison were categorized into three main GO classes (Appendix A). Within the Molecular Function (MF) category, the significantly enriched subclasses included catalytic activity and binding. In the Biological Process (BP) category, the predominant subclasses were metabolic process and cellular process. With respect to the Cellular Component (CC) category, the enriched subclasses comprised cellular anatomical entity and protein-containing complex.

To determine the major functional terms enriched in DEGs, the top 50 significantly enriched GO terms were selected for the AR vs. SR comparison (Figure 4A). In the MF category, glucosytransferase activity (GO:0046527, 40 genes) was the term that showed the highest enrichment. In the BP category, the three most significantly enriched terms were photosynthesis (GO:0015979, 42 genes), phenylpropanoid metabolic process (GO: 0009698, 39 genes), and phenylpropanoid biosynthetic process (GO:0009699, 36 genes). In the CC category, the main enriched terms were categorized into extracellular space (GO:0005615, 26 genes) and photosystem (GO:0009521, 17 genes).

According to the KEGG analysis (Figure 4B), the top 50 KEGG-enriched pathways can be classified into five major functional groups, including metabolism, environmental information processing, genetic information processing, and so on. Within the metabolism group, the most significantly enriched pathways included carbon metabolism (ko01200, 33 genes), phenylpropanoid biosynthesis (ko00940, 29 genes) (Appendix A), and starch and sucrose metabolism (ko00500, 28 genes). In the category of environmental information processing, two pathways were identified as enriched: the MAPK signaling pathway (ko04016, 80 genes) and plant hormone signal transduction (ko04075, 73 genes). Moreover, in terms of cellular processes, the peroxisome pathway (ko04146, 20 genes) was discovered to be enriched. With regard to genetic information processing, the mismatch repair pathway (ko03430, 8 genes) was also enriched. Phenylalanine serves as a crucial precursor for the synthesis of anthocyanins and other flavonoids. Differential expression of genes involved in the phenylpropanoid biosynthetic pathway may affect anthocyanin biosynthesis in the flowers of AR and SR.

### 3.4. Expression of DEGs Related to Flavonoid Biosynthesis and Phenylpropanoid Biosynthesis

The expression profiles of DEGs associated with the biosynthetic pathways of flavonoids and phenylpropanoids were analyzed. Forty DEGs were mapped to these two pathways (Appendix A). Among them, 29 DEGs involved in the phenylpropanoid biosynthetic pathway were identified, including 17 upregulated and 12 downregulated genes in AR. Furthermore, 11 DEGs were involved in the flavonoid biosynthetic pathway. Of these, 4 DEGs related to flavone and flavonol biosynthesis were upregulated, while 4 DEGs involved in anthocyanin biosynthesis were downregulated in AR. Additionally, three DEGs were associated with isoflavonoid biosynthesis. These DEGs may be related to the color differences between AR and SR.

### 3.5. Expression Analysis of Transcription Factors (TFs)

Previous studies have shown that MYB, bHLH, and WD40 family members frequently assemble into MBW (MYB-bHLH-WD40) protein complexes to collaboratively control anthocyanin synthesis [28,29,30]. In this study, DEGs encoding *TFs* were identified between SR and its mutant AR. These differentially expressed *TFs* are summarized in Appendix A, and 105 DEGs were detected in total, as illustrated in Figure 5.

The DEGs included 15 *MYBs*, 11 *bHLHs*, 11 *HBs*, 7 *NACs*, 7 *WRKYs*, 7 *C2H2s*, 6 *AP2/ERFs*, 4 *MADSs*, and 2 *bZIPs*, among others. *Chr1g0360311* (*MYB*) was highly expressed in the petals of SR. Compared to SR, it was significantly downregulated in the petals of AR, showing a more than 40-fold decrease in expression level. In contrast, the expression levels of three MYB genes (*Chr6g0308731*, *Chr5g0039751*, and *Chr7g0177631*) were markedly upregulated in AR petals. Furthermore, *Chr4g0440111* (*bHLH*) and *Chr2g0128051* (*HB-WOX*) were significantly downregulated in AR compared to SR. Additionally, *Chr5g0012431* (*MADS*) was significantly upregulated in AR, exhibiting a more than 16-fold increase in expression. *Chr5g0045961* (*NAC*), *Chr6g0309391* (*C2H2*), *Chr4g0436911* (*bZIP*), and *Chr7g0202671* (*WRKY*) were also upregulated in AR. *Chr7g0231501* (*AP2/ERF*) exhibited significant downregulation, whereas *Chr5g0032721* (*AP2/ERF*) showed upregulation in AR. Overall, these differentially expressed *TFs* are promising candidates for regulating flavonoid biosynthesis and environmental stress responses in roses.

### 3.6. Expression Analysis of GSTs, MATEs and ABCCs

Previous studies have demonstrated that members of the glutathione S-transferase (GST) family, the multidrug and toxic compound extrusion (MATE) family, and the ATP-binding cassette transporter subfamily C (ABCC) contribute significantly to the vacuolar sequestration of flavonoids from the cytoplasm [26,27]. In this study, a total of 42 *GSTs*, 17 *ABCCs*, and 46 *MATEs* were identified as being expressed in the petals of AR and SR. Compared with SR, five *GSTs* and four *ABCCs* were significantly downregulated in AR (Appendix A). For instance, *Chr3g0468161* (*GST*) and *Chr2g0142771* (*ABCC*) showed approximately 2.5-fold downregulation in AR petals. Among the 12 identified *MATEs*, eight were downregulated and four were upregulated in AR (Appendix A). Therefore, the higher expression levels of these *GSTs*, *ABCCs*, and *MATEs* in SR petals suggest that they may contribute to robust anthocyanin sequestration in petal vacuoles.

### 3.7. Expression of Genes Related to the Biosynthetic Pathway of Flavonoids

The expression patterns of genes associated with the biosynthetic pathways of flavonoids were examined in the petals of AR and SR. A total of 37 genes were found to be involved in this pathway (Figure 6, Appendix A). Transcriptomic comparisons between SR and AR revealed that three flavonoid-related genes exhibited significant differential expression. At the naringenin formation stage, two chalcone synthase (CHS) genes—*Chr1g0316441* and *Chr1g0316451*—were markedly downregulated in AR compared to SR. At the anthocyanin formation stage, *Chr7g0199941* (anthocyanidin synthase, ANS) and *Chr2g0153231* (anthocyanidin 3*-O-*glucosyltransferase, 3GT) were also significantly downregulated in AR. The differential expression of these genes participating in flavonoid biosynthesis may be crucial factors leading to the marked reduction in anthocyanin content observed in AR petals (Table 2).

Furthermore, within the phenylpropanoid biosynthetic pathway (Figure 6, Appendix A), caffeic acid was catalyzed by caffeic acid 3*-O-*methyltransferase (COMT) to form ferulic acid. Three *COMT* genes (*Chr1g0382961*, *Chr1g0367691*, and *Chr6g0296031*) were significantly upregulated in AR. Caffeoyl-CoA was converted into feruloyl-CoA through the catalytic action of caffeoyl-CoA O-methyltransferase (CCoAOMT). Notably, *Chr2g0092641* showed marked upregulation in AR. These findings suggest that the upregulation of the aforementioned genes may contribute to a reduction in naringenin chalcone levels in AR.

### 3.8. qRT-PCR Validation of Gene Expression Patterns

To further verify the RNA-seq results, three DEGs associated with flavonoid biosynthetic structural genes and five TFs were selected. Subsequently, quantitative real-time PCR (qRT-PCR) was used to determine their expression levels in the petals of AR and SR at the blooming stage. The primer sequences for these genes are provided in Appendix A. The results confirmed that the flavonoid biosynthetic genes and TFs, including *CHS* (*Chr1g0316451*), *ANS* (*Chr7g0199941*), *MYB* (*Chr1g0360311*), and *AP2/ERF* (*Chr7g0231501*), were downregulated, while *COMT* (*Chr1g0382961*), *WRKY* (*Chr7g0202671*), *bZIP* (*Chr4g0436911*), and *NAC* (*Chr5g0034761*) were upregulated in the petals of AR (Figure 7). Overall, our findings showed that a high degree of consistency was detected between the qRT-PCR and RNA-seq data, validating the reliability of the RNA-seq results and the conducted gene expression analysis.

### 3.9. Network Analysis

To elucidate the regulatory mechanisms underlying anthocyanin biosynthesis, DAMs and DEGs associated with flavonoid pathways were screened from AR and SR based on Pearson correlation analysis. This analytical approach was employed to identify significant correlations between gene expression and metabolite accumulation (Appendix A).

As shown in Figure 8, a network composed of 33 nodes and 262 edges was observed (Appendix A). Within this network, 14 DAMs involved in the flavonoid biosynthetic pathway acted as hub nodes. Among these correlations, 148 pairs showed positive correlations, while 114 pairs presented negative correlations. In particular, *CHSs* (*Chr1g0316441* and *Chr1g0316451*), *ANS* (*Chr7g0199941*), *3GT* (*Chr2g0153231*), *MYB* (*Chr1g0360311*), *bHLH* (*Chr4g0440111*), *GST* (*Chr3g0468161*), and *ABCC* (*Chr2g0142761*) exhibited strong positive correlations with taxifolin, quercetin, and 11 anthocyanins, while showing strong negative correlations with kaempferol. In contrast, *COMT* (*Chr1g0382961*), *CCoAOMT* (*Chr2g0092641*), *MATEs* (*Chr7g0185081*, *Chr2g0159351,* and *Chr2g0112071*), *MYB* (*Chr6g0308731*), and *MADS* (*Chr5g0012431*) showed strong negative correlations with taxifolin, quercetin, and the 11 anthocyanins.

## 4. Discussion

### 4.1. Effects of Flavonoid Content in Rose Petals of SR and AR on Flower Coloration

In this study, 277 DAMs were identified in total between the SR and its bud mutant AR (Table 2 and Appendix A). Our results revealed that quercetin, kaempferol and their multiple derivatives were detected in AR and SR. Early studies have demonstrated that quercetin and kaempferol are the main aglycones of flavonols in rose-species petals [44,45,46]. This study further validates these findings in SR and AR. Moreover, the petals of SR and AR mainly contain three kinds of anthocyanidins: cyanidin, pelargonidin, and peonidin. The findings are consistent with the perspectives presented by Wen et al. [22].

Furthermore, it has been previously reported that high levels of peonidin 3,5*-O-*diglucoside (Pn3G5G) + cyanidin 3,5*-O-*diglucoside (Cy3G5G), and Cy3G5G/ Pn3G5G, are crucial factors for the red coloration of petals in *Rosa rugosa* [47]. The findings of our study indicate that significant amounts of Cy3G5G and Pn3G5G were detected in the red petals of SR, in contrast to AR. Additionally, small quantities of anthocyanins were also identified in the white flowers of AR. The findings align with the study results of Zan et al. [47]. Compared to SR, the contents of anthocyanins and proanthocyanidins in AR decreased significantly, which leads to the white flower phenotype of AR. Most flavonols and flavones exhibited higher contents in AR than in SR. This phenomenon may be attributed to a blockage in the anthocyanin synthetic pathway, which diverts metabolic flux into other related pathways. Additionally, anthocyanins are capable of safeguarding plants from fungal infections, protecting them from high-intensity light, and helping them respond to both biotic and abiotic stresses [9,10,11,12,48,49]. We have also observed that SR plants showed a greater resistance to fungal infections compared to AR plants.

### 4.2. Anthocyanin-Related Genes Affect Flower Coloration in AR and SR

Anthocyanins play a crucial role as one of the primary elements responsible for producing a wide range of colors, from orange/red to violet/blue. They are synthesized in the cytosol and sequestered within the vacuoles of plant cells [4,50]. The biosynthesis of anthocyanins is a complex process that encompasses various biochemical pathways and interconnected gene expression networks. In our research, we analyzed DEGs between SR and its color mutant AR, and identified the differentially expressed structural genes involved in the biosynthesis of flavonoids and phenylpropanoids.

CHS serves as the crucial rate-limiting enzyme in the initial stage of flavonoid biosynthesis. It promotes the formation of the basic carbon skeleton of anthocyanins [51]. Previous studies have shown that silencing *CHS* leads to a significant decrease in anthocyanin content, resulting in lighter fruit color in tomatoes [52]. Conversely, overexpression of *VdCHS2* promoted anthocyanin biosynthesis, which contributed to the establishment of a high-yield anthocyanin cell line named OE1 in *Vitis davidii* [53]. In the present study, it is deduced that the low expression levels of *CHS* in AR petals may cause a shortage of precursor compounds, thereby contributing to the decreased anthocyanin content.

Anthocyanin synthase acts as the rate-limiting enzyme in the later stages of anthocyanin biosynthesis, promoting the transformation of colorless precursors into colored anthocyanins [41]. Previous studies have verified that the *ANS* gene functions as a key enzyme gene responsible for anthocyanin accumulation in various plant tissues, including perilla, *Arabidopsis*, pansy, pomegranate ‘Chuju’, and potato, thereby playing a crucial role in pigment accumulation across different species [54,55,56,57,58,59]. In apples, the downregulation of *ANS* expression results in a decrease in anthocyanin biosynthesis [60]. In begonias, *McANS* is vital for petal pigmentation, with its transcriptional activity exerting a substantial influence on the reddish color of blossoms [61]. In the present study, the expression of *ANS* (*Chr7g0199941*) was found to be downregulated in AR petals. This downregulation may lead to the observed white flower coloration.

Among structural genes, UFGT (UDP glucose: flavonoid glucosyltransferase) participates in the late stage of anthocyanin biosynthesis and plays a vital role. Kobayashi et al. [62] suggested that the phenotypic alteration from white to red in grape sports might be due to a mutation in a regulatory gene that governs the expression of *UFGT*. Additionally, *PavMYB.C2* upregulates the expression level of the anthocyanin structural gene *UFGT*, resulting in anthocyanin accumulation in cherry fruit [63]. RhGT74F2 was validated, confirming their role in anthocyanin accumulation in the ‘Rhapsody in Blue’ rose [43]. In our study, the expression of *3GT* (*Chr2g0153231*) was significantly downregulated in AR petals, suggesting that it might contribute to the alteration of petal color. The findings indicated a substantial reduction in the expression levels of *CHS*, *ANS*, and *3GT*, accompanied by a remarkable increase in the expression levels of *CCoAOMT* and *COMT* in AR petals. Such changes could directly lead to a decrease in the contents of anthocyanins and proanthocyanidins, thereby resulting in the white flower phenotype of AR.

In plants, the biosynthesis of anthocyanins is predominantly regulated by the MYB-bHLH-WD40 (MBW) protein complex [33,34,35,36]. Moreover, MYB transcription factors act as crucial regulators in the biosynthetic pathway of anthocyanins [35,36,37]. Early studies have demonstrated that *VvMYB5a* and *VvMYB5b* participate in regulating anthocyanin accumulation during grape berry ripening by enhancing the expression of *ANS* [64]. In roses, *RcMYB1* and *RcMYB114* are involved in the regulation of anthocyanin biosynthesis [36,37]. Similarly, *RcMYB114a*, *RcMYB114b*, *RcMYB114c*, and *RcMYB114d*, together with the *RcbHLH* gene, result in the accumulation of anthocyanins, and produce red coloration [65]. Additionally, it has been shown that in transgenic tobacco, the overexpression of *MdMYB3* can upregulate the expression levels of *CHS* and *CHI* genes, promoting anthocyanin accumulation in flowers [66]. The homologous gene of *MdMYB10*, *MdMYB110*, also regulates anthocyanin biosynthesis in apple flesh. It likely participates in the formation of the MBW complex, which subsequently activates *CHS* expression [67]. Our results revealed that *Chr1g0360311* (*MYB*) exhibited strong positive correlations with taxifolin, quercetin, and 11 anthocyanins, and showed significant downregulation in AR petals. This gene shares high sequence similarity with *AtMYB3*, a transcription factor reported to participate in the regulation of anthocyanin biosynthesis [68]. These results indicate that *Chr1g0360311* might act as a crucial regulatory factor for structural genes related to anthocyanin biosynthesis, thus affecting flower color variation.

### 4.3. Downregulation of GST, ABCC and MATE Genes May Decrease Anthocyanin Storage in the Vacuoles of AR Flowers

The biosynthesis of flavonoids occurs on the cytoplasmic side of the endoplasmic reticulum; however, various flavonoids ultimately accumulate in the vacuole [32]. Earlier studies have indicated that glutathione S-transferases, membrane transporters, and vesicle trafficking constitute the three primary mechanisms involved in flavonoid transport in plants [31]. Members of the GST, ABCC, and MATE transporter families are critical to the sequestration of flavonoids from the cytoplasm into vacuoles [31,32].

In the Arabidopsis *tt19* loss-of-function mutant, the accumulation of anthocyanins in vegetative tissues and the content of brown pigments in the seed coat were significantly decreased, indicating that *TT19* is critical for the transport of anthocyanins and proanthocyanidins [69]. Previous studies have demonstrated that *RcGSTF2* in rose and *GhTT19* in upland cotton are involved in anthocyanin transport [70,71]. Moreover, a study identified allelic variations in the *GhTT19* promoter across different varieties, which were correlated with differences in petal coloration [71]. The *GST* gene family, including *ScGST3* in *Senecio cruentus*, *LhGST* in *Lilium*, *RAP* in strawberry, *MrGST1* in Chinese bayberry, and *PpGST1* in peach, has been shown to be specifically involved in anthocyanin transport, rather than in proanthocyanidin transport [72,73,74,75,76]. In contrast, *AcGST1* in *Actinidia chinensis* and *RsGST1* in radish are not only involved in anthocyanin transport but also contribute to the vacuolar sequestration of proanthocyanidins [77,78]. Additionally, Perez-Diaz et al. [79] reported that grape GSTs can bind not only anthocyanins but also proanthocyanidins and monomeric flavonols, indicating that the substrate specificity of GST proteins may differ across plant species. In this study, five *GST* genes were identified as being downregulated in AR petals, suggesting that their reduced expression may result in decreased anthocyanin sequestration in the vacuoles of AR floral tissues.

The subfamily of multidrug resistance-associated proteins (MRP/ABCC) belongs to the ATP-binding cassette (ABC) superfamily [80]. This group of proteins is known to participate in the transport of flavonoids [31]. According to Francisco et al. [81], *VvABCC1* in grapevine participates in the transport of glucosylated anthocyanidins. Furthermore, AtABCC2—a homolog of ZmZRP3 in *Arabidopsis thaliana*—plays a dual role in the vacuolar transport of anthocyanins as well as flavonoids and flavonols [82]. Transient overexpression of *PpABCC1* in peach (*Prunus persica*) led to a remarkable increase in the accumulation of anthocyanins in both tobacco leaves and peach fruits. In contrast, virus-induced gene silencing of *PpABCC1* resulted in a marked reduction in anthocyanin levels, indicating that *PpABCC1* is crucial for anthocyanin accumulation in peach [83]. Our findings suggest that the four *ABCC* genes were downregulated in AR petals, implying their potential involvement in anthocyanin transport.

In plants, MATE proteins are widely involved in various biologically active processes, including the accumulation of secondary metabolites, the excretion of xenobiotics, plant hormone signal transduction, aluminum tolerance, and disease resistance [84]. In *Arabidopsis*, AtTT12 can transport two flavonoid compounds: cyanidin 3*-O-*glucoside (Cy3G) and epicatechin 3′*-O-*glucoside (E3′G) [85]. *MtMATE1* in *Medicago truncatula* has been shown to participate in the transport of proanthocyanidins [86]. *MtMATE2* and *PhMATE1* participate in the transport of anthocyanins [87,88]. Moreover, *CaMATE1* plays a critical role in the accumulation of proanthocyanidins and anthocyanins in chickpea flowers and seed coats [89]. Additionally, in soybean, *GmMATE1* and *GmMATE4* function as isoflavone transporters [90,91]. In this study, eight downregulated *MATEs* and four upregulated *MATEs* were identified in AR petals. These *MATE* family members may play an important role in the accumulation of anthocyanins, proanthocyanidins, or other flavonoid compounds in roses.

## 5. Conclusions

In this study, compared with the SR, the contents of anthocyanins and proanthocyanidins exhibited a substantial decrease in the petals of its bud mutant AR. The mutation caused a significant downregulation of crucial structural genes involved in anthocyanin biosynthesis and sequestration. Additionally, there was a notable up- regulation of specific genes related to the phenylpropanoid biosynthetic pathway. *MYB* (*Chr1g0360311*) showed a significant difference in expression between AR and SR. As a result, the accumulation of anthocyanins and proanthocyanidins was significantly reduced, leading to white flowers in AR. The functions of candidate structural genes and specific *MYB* transcription factors will be explored in subsequent research.

## Figures and Tables

**Figure 1 biology-14-01337-f001:**
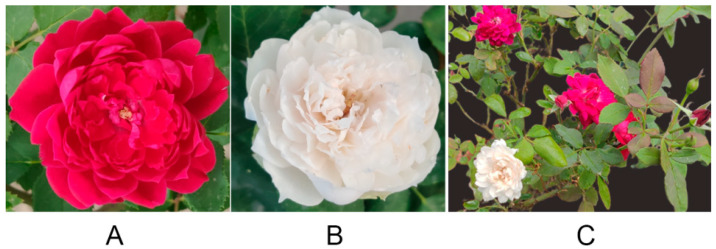
Phenotypic characteristics of the ‘Silk Road’ and its bud mutant ‘Arctic Road’. (**A**) ‘Silk Road’. (**B**) ‘Arctic Road’. (**C**) The natural mutant from the cultivar ‘Silk Road’.

**Figure 2 biology-14-01337-f002:**
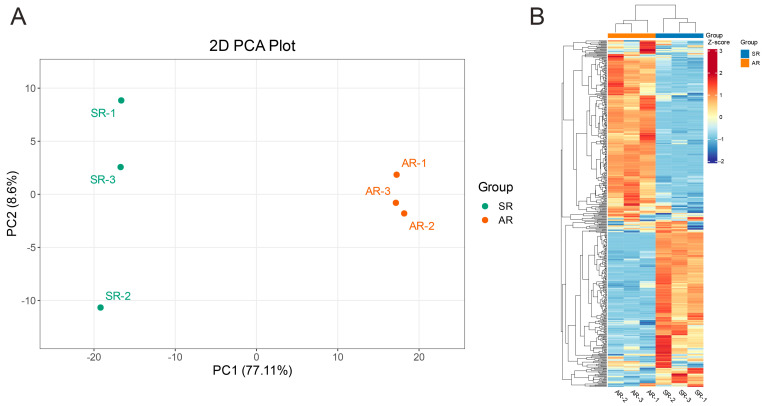
Comparative analysis of metabolites extracted from the petals of AR and SR. (**A**) PCA score plots for AR and SR. (**B**) A heatmap illustrating 479 metabolites identified in rose petals. The color scale, ranging from red to green, represents the normalized metabolite contents calculated by the row Z-score.

**Figure 3 biology-14-01337-f003:**
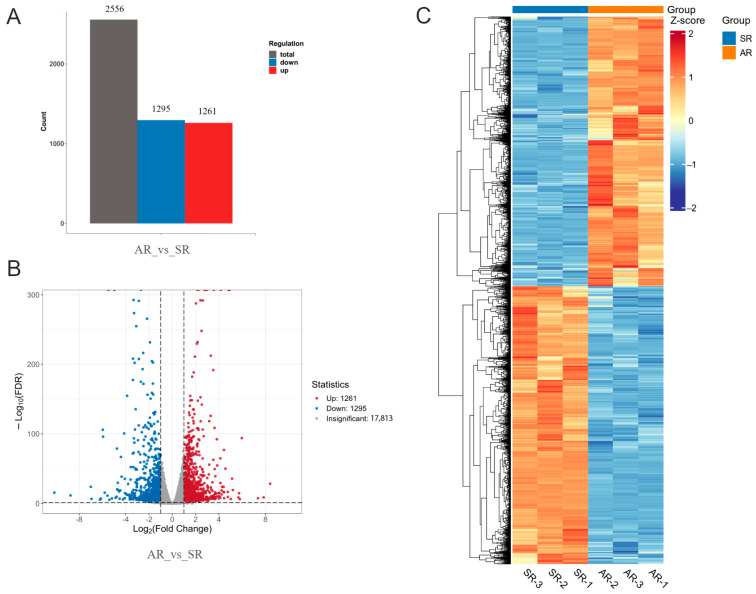
Comparative analysis of DEGs in the petals of AR and SR. (**A**) The number of DEGs between AR and SR. (**B**) Volcano plots of DEGs in AR vs. SR. Red dots represent upregulated DEGs, blue dots indicate downregulated DEGs, and grey dots denote non-differentially expressed transcripts. (**C**) A heatmap of 2556 DEGs from the rose petals of AR and SR. The color scale, ranging from red to blue, represents the normalized transcripts calculated by the row Z-score.

**Figure 4 biology-14-01337-f004:**
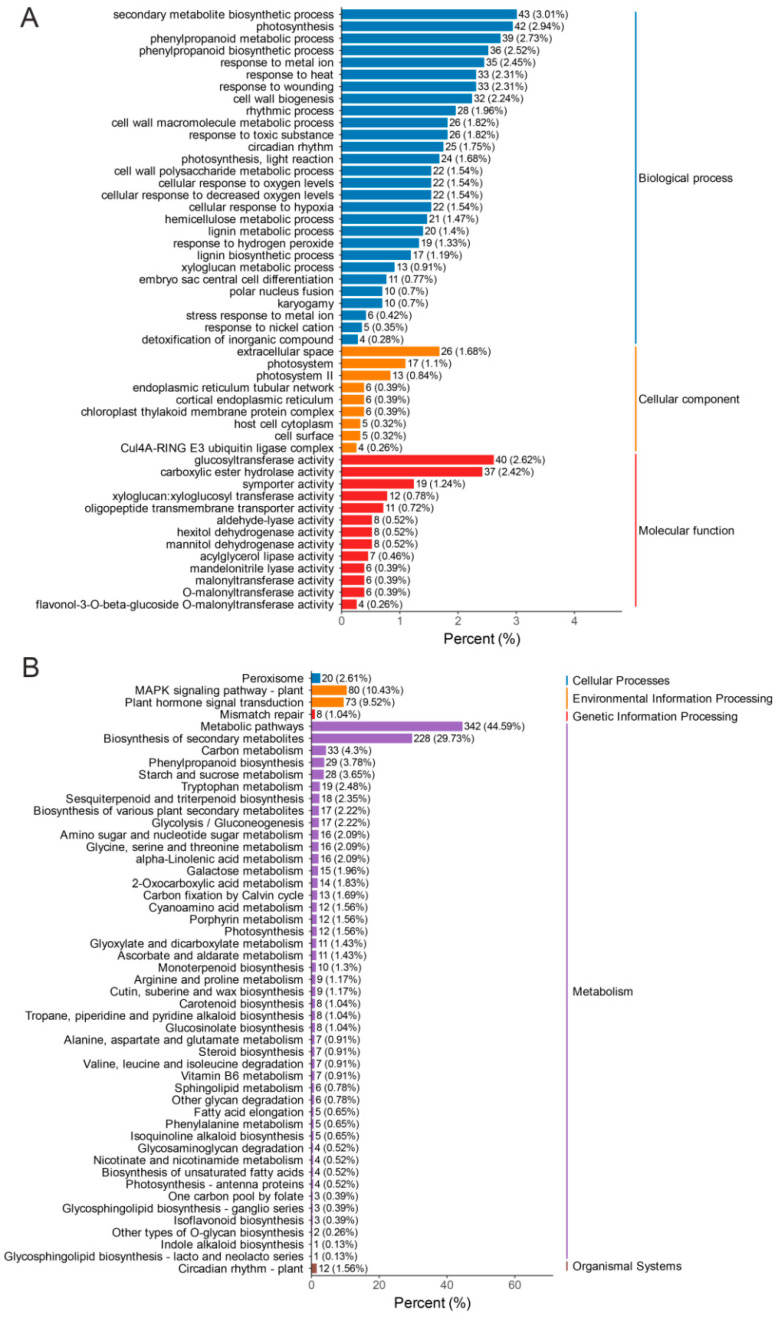
Transcriptome-based enrichment analysis of DEGs in AR vs. SR. (**A**) The top 50 enriched GO terms. The numbers following the bars denote the number of DEGs annotated to each entry. The ratios in parentheses signify the proportion of DEGs annotated to the GO entry relative to the total number of annotated DEGs. The labels on the far right indicate the category to which the GO entry belongs. (**B**) The top 50 enriched KEGG pathways. The numbers following the bars denote the quantity of DEGs annotated to the pathway. The ratios in parentheses signify the proportion of DEGs annotated to the pathway relative to the total number of annotated DEGs. The labels on the far right represent the classification of the KEGG pathways.

**Figure 5 biology-14-01337-f005:**
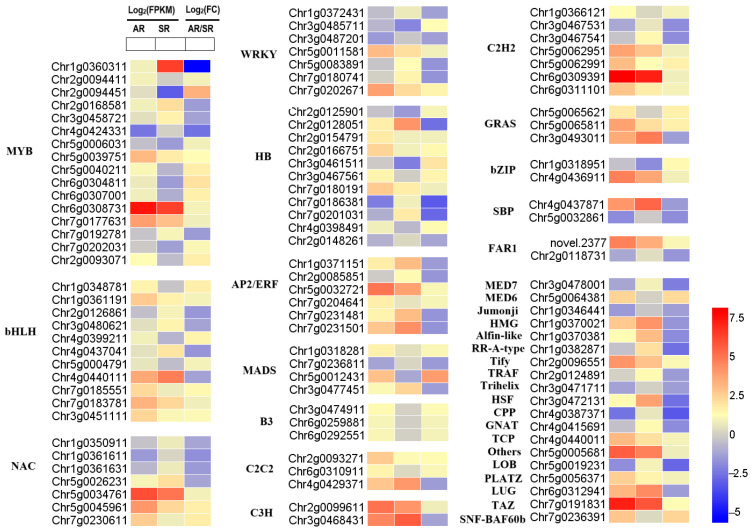
Heatmaps displaying DEGs based on log_2_(FPKM) and log_2_(FC) values of *TFs* in the AR vs. SR comparison. The color scale represents log_2_(FPKM) and log_2_(FC) values.

**Figure 6 biology-14-01337-f006:**
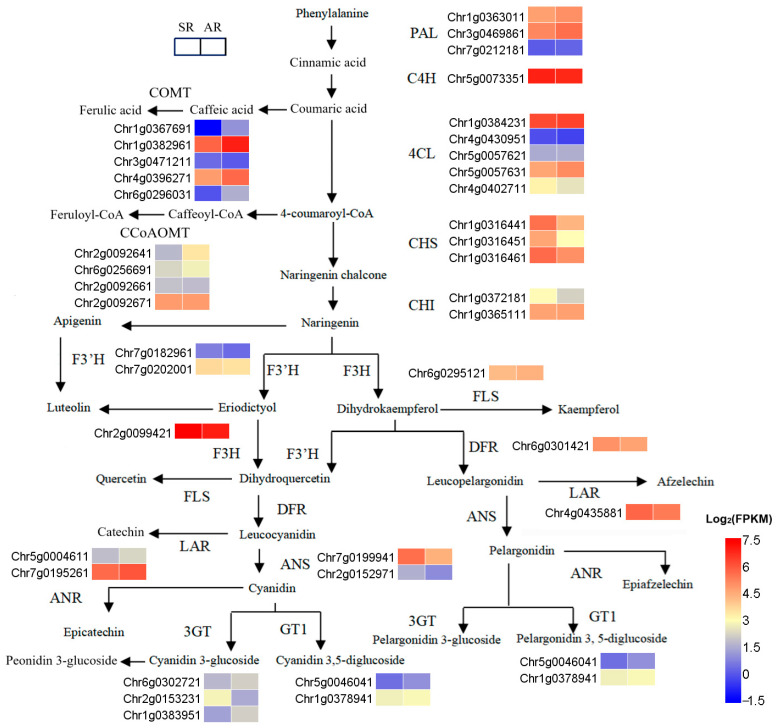
Biosynthetic pathway of flavonoids. Heatmaps displaying the expression levels of genes involved in flavonoid biosynthesis in the petals of AR and SR. PAL, phenylalanine ammonialyase; C4H, cinnamate-4-hydroxylase; COMT, caffeic acid 3*-O-*methyltransferase; 4CL, 4-coumarate-CoA ligase; CHS, chalcone synthase; CCoAOMT, caffeoyl-CoA O-methyltransferase; CHI, chalcone isomerase; F3′H, flavanone 3′-hydroxylase; F3H, flavanone 3-hydroxylase; FLS, flavonol synthase; DFR, dihydroflavonol 4-reductase; ANS, anthocyanin synthase; ANR, anthocyanin reductase; GT1, anthocyanidin 5, 3*-O-*glucosyltransferase; LAR, leucoanthocyanin reductase; 3GT, anthocyanidin 3*-O-*glucosyltransferase. The color scale represents log_2_(FPKM) values.

**Figure 7 biology-14-01337-f007:**
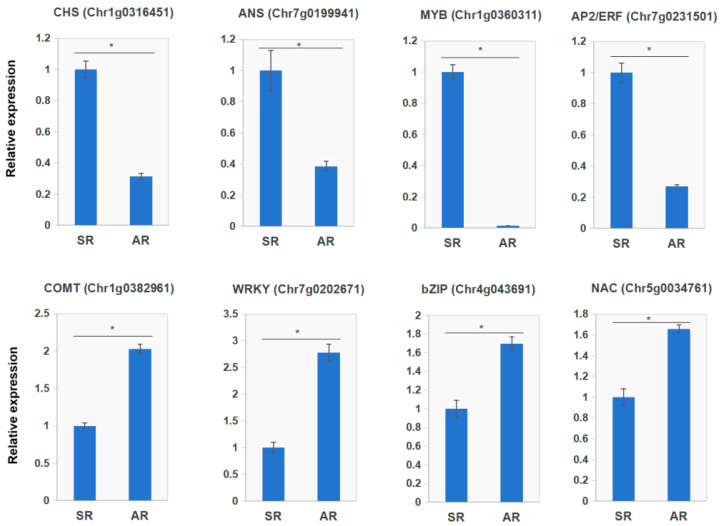
qRT-PCR validation of transcripts from AR and SR. Asterisks (*) indicate that the transcript levels of genes (*n* = 3, ±SD) are significantly different between AR and SR at *p* < 0.05.

**Figure 8 biology-14-01337-f008:**
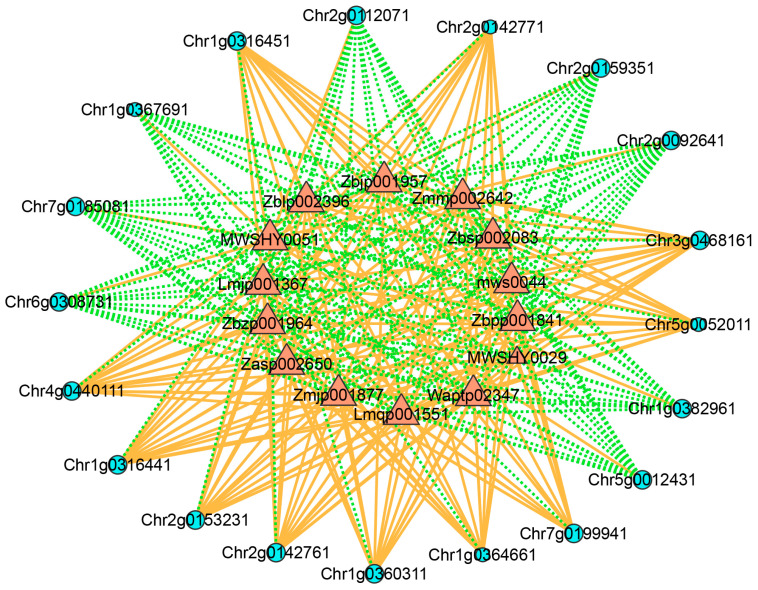
Diagram of gene–metabolite correlation networks. Red triangles denote metabolites, while blue circles represent genes. Positive and negative correlations are signified by orange solid lines and green dashed lines, respectively. MWSHY0029, quercetin; mws0044, taxifolin; MWSHY0051, kaempferol; Zbjp001957, cyanidin 3,5*-O-*diglucoside; Zbsp002083, pelargonin 3,5*-O-*diglucoside; Waptp02347, peonidin 3*-O-*sophoroside; Zmjp001877, cyanidin 3*-O-*beta-d-sambubioside; Zblp002396, peonidin 3*-O-*glucoside; Lmqp001551, cyanidin 3,3′,5-tri*-O-*glucoside; Zbzp001964, cyanidin-3-di-glucoside-5-glucoside; Zasp002650, cyanidin-3*-O-*galloyl-galactoside; Zmmp002642, cyanidin-3*-O-*(6″*-O-*feruloyl) glucoside; Lmjp001367, cyanidin 3*-O-*(beta-D-xylosyl- (1→2)-beta-d-galactoside); Zbpp001841, peonidin-3,5*-O-*diglucoside.

**Table 1 biology-14-01337-t001:** Statistical data of differentially accumulated metabolites in the petals of AR and SR.

Class	Number (AR vs. SR)	
	Up	Down
Anthocyanidins	0	11
Aurones	3	3
Chalcones	5	4
Flavonols	88	38
Flavones	41	27
Isoflavones	1	3
Flavanols	5	11
Flavanones	3	6
Flavanonols	1	1
Other Flavonoids	2	5
Proanthocyanidins	0	11
Tannins	4	4
Total	153	124

**Table 2 biology-14-01337-t002:** Differentially accumulated anthocyanidins and proanthocyanidins in the petals of AR and SR.

Class	Metabolite	Content		Log2(FC)	VIP
		AR_Mean	SR_Mean		
Anthocyanins	Cyanidin 3,5*-O-*diglucoside	2.47 × 10^5^	7.17 × 10^6^	−4.86	1.140
	Cyanidin 3*-O-*beta-D-sambubioside	8.23 × 10^5^	6.67 × 10^6^	−3.02	1.136
	Cyanidin-3*-O-*galloyl-galactoside	1.74 × 10^5^	2.18 × 10^6^	−3.64	1.137
	Cyanidin-3-diglucoside-5-glucoside	3.60 × 10^3^	3.67 × 10^4^	−3.35	1.139
	Cyanidin 3,3′,5-tri*-O-*glucoside	1.33 × 10^4^	1.00 × 10^5^	−2.91	1.142
	Cyanidin-3*-O-*(6″*-O-*feruloyl)glucoside	6.18 × 10^4^	5.79 × 10^5^	−3.23	1.142
	Cyanidin 3*-O-*(beta-D-xylosyl-(1→2)-beta-D-galactoside)	5.62 × 10^4^	7.24 × 10^6^	−7.01	1.152
	Pelargonidin 3,5-di-beta-D-glucoside	8.00 × 10^4^	6.32 × 10^5^	−2.98	1.137
	Peonidin 3*-O-*sophoroside	4.55 × 10^4^	3.67 × 10^6^	−6.33	1.141
	Peonidin 3*-O-*glucoside	1.17 × 10^5^	7.46 × 10^6^	−6.00	1.142
	Peonidin-3,5*-O-*diglucoside	4.49 × 10^4^	3.38 × 10^6^	−6.23	1.141
Proanthocyanidins	Procyanidin B8	9.15 × 10^5^	2.41 × 10^6^	−1.40	1.141
	Procyanidin A4	2.14 × 10^5^	1.73 × 10^6^	−3.02	1.142
	Procyanidin C2	3.71 × 10^4^	1.42 × 10^5^	−1.94	1.122
	procyanidin B4 3*-O-*gallate	4.72 × 10^4^	1.22 × 10^5^	−1.37	1.126
	Cinnamtannin A1	1.06 × 10^4^	4.41 × 10^4^	−2.05	1.112
	Proanthocyanidin A2	1.53 × 10^4^	2.02 × 10^5^	−3.72	1.085
	3-galloylProcyanidin B1	3.43 × 10^4^	1.24 × 10^5^	−1.86	1.044
	Procyanidin A1	5.42 × 10^3^	4.31 × 10^4^	−2.99	1.125
	2α,3α-Epoxy-5,7,3′,4′-tetrahydroxyflavan-(4β→8)-catechin	3.80 × 10^4^	3.97 × 10^5^	−3.38	1.142
	2α,3α-Epoxy-5,7,3′,4′-tetrahydroxyflavan-(4β→8)-epicatechin	2.74 × 10^4^	2.45 × 10^5^	−3.16	1.111
	9,10-Dihydro-10-(4-hydroxyphenyl)-pyrano [2,3-h]epicatechin-8-one gallate	1.84 × 10^5^	2.70 × 10^6^	−3.88	1.139

## Data Availability

The original contributions presented in this study are included in the article/Appendix A. The RNA-seq datasets generated during this study are publicly available in the NCBI Sequence Read Archive (SRA) under accession number PRJNA1312512. Further inquiries can be directed to the corresponding author.

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
