# Peer review of "Integrated Metabolomic and Transcriptomic Analyses of the Flavonoid Biosynthetic Pathway in Relation to Color Mutation in Roses"

_biology, 2025, doi:10.3390/biology14101337_

Round 1

Reviewer 1 Report

Comments and Suggestions for Authors

This manuscript presents a metabolite and RNA-seq comparison between two rose cultivars, Silk Road and Arctic Road, which differ in flower color. The authors compare levels of flavonoids and genes related to anthocyanin biogenesis between the cultivars. The authors report many changes in gene expression and metabolites and comment on the implications of those changes for production of anthocyanin and other metabolites. This paper would be of interest to readers interested in roses, and/or genes and flavonoids related to flower color.

General comments:

From the limited description in the methods, it’s unclear how close the growing and sampling conditions were between these two cultivars. The number of DEGs is higher than would be expected for a mutation affecting anthocyanin alone. Can you comment on any differences between growing conditions or sampling time which might contribute to differential gene expression. Both global transcription and metabolite levels differ significantly throughout the day-night cycle.

Could you use the RNA-seq reads generated in this experiment to look for changes in the amino acid sequences of the anthocyanin related genes highlighted in this study to identify potentially non-functional alleles in the AR cultivar?

Specific comments:

110: Is anything known about what type of mutation distinguishes these cultivars? Are they fairly isogenic, or are there other differences between the backgrounds besides the color? Are the changes in flower color likely caused by a single mutation, or are there many mutations between the cultivars? Do observable differences between the cultivars extend beyond flower color?

112: Where all the samples collected the same day? What time of day were they collected?

134: Can you provide more details about the RNA extraction?

153: how closely related are these cultivars to the reference genome? [this may be better addressed in the results section]

247: Knowing how closely related these cultivars are to the reference would help give context to this mapping rate. [see previous comment]

250: Is 39,201 the number of genes with FPKM greater than 1 or the total number of genes in the genome. If it’s not the total number of genes in the genome, could you add that number to give some context?

254: Where does the 20,369 number come from? How is it related to the 39,201 number?

438-453: Some additional hypothesizing about specific aspects of anthocyanin production as sell as potentially causative genes would be welcome here.  

511: extra space in line.

Figures:

Figure 2: The colors on this figure are not friendly to those with color vision deficiencies. Matching the colors used in the heat map in Figure 3 would make sense. In panel B, an additional column representing the VIP score for each metabolite, could be interesting since 716 metabolites are shown, but only 277 are significantly changed.

Figure 4: Are all of these statistically significantly over-represented? If so, could you add p-values to this figure?

Figure 5: The colors on this figure are not friendly to those with color vision deficiencies. Are all the genes shown in this figure significantly different? If so, could you include p-values? It would also be nice to see a representation of the fold-change between the two cultivars. You could have 4 columns for each gene with the third column representing fold change, and the fourth representing padj.

Figure 6: See all comments regarding figure 5.

Figure 7: The legend should read “transcript levels” not “transcription levels.” qRT-PCR measures levels of RNA present not rates of transcription.

Figure 8: The colors on this figure are not friendly to those with color vision deficiencies. It does help that the green lines are also dashed, but a different color selection would be better.

Tables:

Table 2: VIP scores would be a nice addition to this table. 

Comments on the Quality of English Language

There are some grammatical errors that make some sections hard to follow. 

Reviewer 2 Report

Comments and Suggestions for Authors

The authors of the manuscript “ Integrated metabolomic and transcriptomic analyses of the flavonoid biosynthetic pathway in relation to color mutation in roses” revealed that the study was conducted to investigate the mechanisms underlying the formation of rose color. The red petals from the rose cultivar 'Silk Road' (SR) and the white petals from its color mutant 'Arctic Road' (AR) were investigated. Transcriptomic and metabolomic analyses were utilized to identify the  crucial genes and metabolites associated with the biosynthesis of flavonoids. The MYB gene (Chr1g0360311) may serve as a key regulator in anthocyanin biosynthesis.

The results of this manuscript are interesting. However, authors need to address the following issues:

  1. The background section focuses solely on relevant knowledge concerning the pathways involved in petal colour formation. Previous studies have explored the mechanisms underlying rose colouration formation; however, the background section of this manuscript lacks an introduction to these mechanisms. Please include relevant research advances concerning rose colours.

  1. Why was petal material from only one stage selected for the experimental materials section? Are the colours of rose petals identical during the bud stage, peak bloom, and wilting stages?

  1. The discussion section lack comparison and discussion with previous studies on the mechanism of rose colour formation.

Reviewer 3 Report

Comments and Suggestions for Authors

The manuscript titled “Integrated metabolomic and transcriptomic analyses of the flavonoid biosynthetic pathway in relation to color mutation in roses” unravel rose cultivar ‘Silk Road’ and its color mutant ‘Arctic Road’ genetic and metabolomic regulation of colors in petals. This topic is highlight of modern and fast technologies to understand complexity of genetic regulation and metabolism. The object of this article is unique as both cultivars are particularly close by mutation of inflorescence color. This eliminates stronger genetic variation than contrasting cultivars might have, so this led to ideal study object. The experiment design was flawless. Authors set to investigate 716 flavonoid-related metabolites and over 39 thousand genes. Their significant differences lead to overview of consistent conclusions that could lead to suggestions for key genes and metabolites for regulation of rose petal color. It is closely related to both fundamental understanding and application for breeders who are looking for specific pigmentation traits in new cultivar development.

Overall, the experimental design and manuscript quality is exceptionally great, and the following points are mainly suggestions for improvements.

Methodology – in line 113 it was stated that petal samples gathered from six flowers and then were pooled to make one sample. How much of each bloom were gathered (weight, number of petals, or parts of petals?), what was the total weight of one pool sample? This information would be essential for replicating this methodology for further analysis by using contrasting genotypes or even using it for other ornamental plants.

Results – in line 117 metabolom-ics. This should be edited before publication.

Results – Figure 1. By the logic cultivar photo B of ‘Silk Road’ should be first, then it’s mutant white form ‘Arctic Road’ photo A. Also, why term wild-type is used? I see it was used in discussion and conclusions too. As I started to look for information of cultivar ‘Silk Road’ I came to notice it is not the same color as you used. Here is the example source I found it (https://www.baranoie.com/shop/shopdetail.html?brandcode=000000010797&search=%A5%B7%A5%EB%A5%AF%A5%ED%A1%BC%A5%C9&sort= ). Could you please clarify the differences or highlight this aspect?

Results – lines 240-242 (First two sentences) is solely methodological parts. It should be incorporated in methodology and not repeated several times across the manuscript.

Results – Figure 5 and Figure 6. I would suggest changing DEGs expression color scale to more contrasting colors. Now colors are hard to read with hues and tones, that from the first look does not show strong differences to visualize tendencies on both AR and SR genotypes. However, when looking more closely it is clear that there are differences, but it would be clearer if colors were from red to blue, or midpoint black. There are several approaches to increase visual appearance by incorporating color map adapted for individuals with color vision deficiencies – this would also improve quality of figures for healthy people and create visually contrasting strong tendencies that you wanted to highlight.

Results – Figure 7. Please indicate statistical significance with horizontal line almost connecting compared groups of SR and AR. Then above the line in the middle use asterisks to denote significance levels. Avoid placing asterisks directly on one individual bar as it is more used for checking gene expression significance from control between several treatment groups. In this case it is not good, as you highlight differences between two groups.

Results – Figure 8. Interesting approach to visualize this type of data, but it is hard to read and understand which metabolites and genes form negative or positive correlations. I would suggest for visualization to use Pearson correlation coefficients. If not – try to look for alternative approach to visualize data so it could be clear to see and understand key aspects.
